# Modeling Time Series as Text Sequence: A Frequency-vectorization Transformer for Time Series Forecasting

## Abstract

Time series is an essential type of sequential feature that widely exists in multiple scenarios (e.g., healthcare, weather prediction) and contains valuable information, so various studies have been conducted for modeling time series. Transformer-based models have achieved great success in modeling sequence data, especially text, but fail to understand time series sequences. The reason is that individual data points of time series are hard to utilize because they are numerical values that cannot be tokenized. To address this challenge, we design a frequency-vectorization time series forecasting method in this paper. Different from most previous studies that adopt frequency domain to extract extra features, we propose to utilize frequency spectrum as a common dictionary for tokenizing time series sequences, which converts single time series into frequency units with weights. Then, the vectorized frequency token sequence can be modeled by transformer layers directly for prediction tasks. Furthermore, to align the frequency and the time domains, we introduce two pretraining tasks: time series reconstruction task and maximum position prediction task. Experimental results on multiple datasets demonstrate that our model outperforms existing SOTA models, particularly showing significant improvements in long-term forecasting. Besides, our model exhibits remarkable transferability across various prediction tasks after pretraining.

## 1 Introduction

Time series is a type of widespread sequential data that exists in various domains of our daily lives, such as heart rate and temperature sequences, which contains valuable information. Due to its tremendous potential for extensive applications, substantial research attention has focused on modeling time series in various tasks, including weather forecasting (Kim et al., 2017), electricity consumption budgeting (Barta et al., 2016), and traffic flow prediction (Cai et al., 2020).

In terms of sequence modeling, transformer-based models show immense promise. They have demonstrated remarkable achievements in dealing with various types of sequential data, particularly in text sequence (Brown et al., 2020; Wei et al., 2022) and DNA sequence (Ji et al., 2021). As is illustrated in Figure 1, transformer-based models first split raw data into fundamental units, and then vectorize the units using a common dictionary. Take the text sequence as an example, a text sentence will be tokenized as a token sequence, and then the tokens are mapped into embedding vectors with a predefined dictionary. After that, these vectors are fed into transformer layers for further calculations, which shows powerful performance in various downstream tasks.

However, when it comes to model time series, it encounters a crucial challenge if we want to utilize the powerful transformer models. Unlike text sequences where individual words can serve as tokens with corresponding representations (Mikolov et al., 2013), individual data points within time series lack such inherent semantics. Therefore, transformer architectures cannot be directly applied to model time series data due to the absence of a clear tokenization scheme. An empirical idea is that we can utilize time series as images, i.e., taking several data points as a token, but it fails to capture the dependencies among data points within a patch while they have influence on each other.

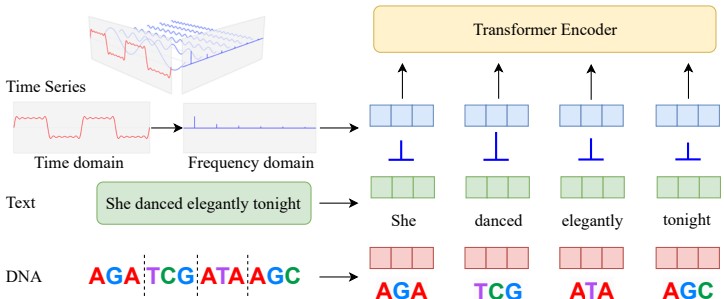

Figure 1: Comparison with other sequential features in utilizing transformer layers for time series. Text sequences are split as tokens, and DNA sequences are tokenized as sub-words. We propose to split the time series into tokens in the frequency domain.

To handle this challenge, we propose a frequency-vectorization method to model time series for the forecasting task demonstrated in Figure 1[1]. Most previous studies transfer time series into the frequency domain for extracting and integrating more features and then transfer back to the time domain (Wu et al., 2021; Zhou et al., 2022b), which demonstrates better forecasting abilities. Different from these methods, we propose to utilize the frequency domain for time series tokenization, where a continuous time series in the time domain will be represented as the amplitudes of a specific set of sinusoid waves with distinct periods in the frequency domain. We can take sinusoid waves as tokens because they have different periods, so the time series becomes a token sequence with weights (amplitudes). The resulting frequency token sequence can then be directly modeled by transformer layers, which integrate all distinct frequency tokens to perform more accurate predictions. Note that if we use the same set of sinusoid waves in the frequency domain for modeling various time series, the token dictionaries are shared to tokenize unseen time series as the common dictionary for text sequences. Thus, this approach empowers us to conduct extensive training on diverse datasets, ultimately enhancing prediction performance across various scenarios for zero-shot transformation.

Through comprehensive experiments conducted on multiple datasets, we demonstrate the superior performance of our proposed model over existing state-of-the-art models. Notably, it exhibits remarkable advancements in long-term forecasting tasks, a critical aspect of time series modeling. Moreover, our model showcases excellent transferability, making it valuable for broader zero-shot applications after pretraining. Our main contributions can be summarized in the following:

- To the best of our knowledge, we first propose to conduct time series tokenization in the frequency domain, so that it is easy to model time series as text sequences.

- We design the **Freq**uency-enhanced **T**ime **S**eries **T**ransformer (FreqTST) for forecasting, which uses the frequencies composing the time series as fundamental units to form a frequency token sequence. Furthermore, we design two new pretraining tasks to align the time and frequency domain, which improves the generalization ability of our model.

- Extensive experimental results demonstrate that our method outperforms the SOTA models on various public datasets, especially on the long-term prediction task. Pretrained on various datasets, FreqTST shows outstanding transferability in zero-shot experiments.

## 2 RELATED WORK

### 2.1 TIME SERIES FORECASTING MODELS

Time series forecasting is an important task that can be applied in various scenarios, so various models are designed to handle it. ARIMA (Pai & Lin, 2005) is an early work that models time series by decomposing and a linear layer, but it performs badly on complex datasets due to its simple structure. Another famous type of method in time series forecasting is LSTM, which predicts the future value in a recurrent way, and thus suffers from computation efficiency (Lai et al., 2018).

---

[1]Partly refer to https://commons.wikimedia.org/wiki/File:Fourier_transform_time_and_frequency_domains.gif

In recent years, Zhou et al. (2021) introduces transformer layers into this task, replacing previous attention mechanism with sparse attention calculation and distilling operation to overcome the efficiency problem of vanilla transformer. Transformer-based models achieve better performances in long-term time series forecasting than previous models, so more and more transformer-based models are designed for better performance (Zerveas et al., 2021; Nie et al., 2023).

Another research trend is the utilization of the frequency domain in forecasting, as time series in the time domain can be transformed into the frequency domain. An intuitive way is to use the frequency domain to extract more features. For instance, Wu et al. (2021) employ the equivalent form of frequency signals in calculations to capture the similarity between sub-sequences. Wu et al. (2023) decompose time series into different periods corresponding to different frequencies, forming a two-dimensional numerical matrix for further calculation. Further studies propose to use the frequency domain for auxiliary computation, which can be helpful in improving the forecasting performances (Deznabi & Fiterau, 2023; Zhao et al., 2023). Zhou et al. (2022b) transform input time series into the frequency domain for computation, randomly sample a fixed number of frequencies in the complex domain for attention calculation, and transform the results back into the time domain. Besides, Zhou et al. (2022a) utilize randomly sampled frequency signals for attention calculation but perform a low-rank approximation to improve efficiency. These techniques help models improve their understanding of time series, which is superior to the previous frequency methods.

Recent works combine representations in the time domain and the frequency domain with distinct combination strategies (Yang & Hong, 2022). These methods contribute to understanding the relationship between the two domains, and some of them also utilize transformer layers to combine these features (Chen et al., 2023b). Besides, Zeng et al. (2023) challenges the effectiveness of transformer-based models, using a linear layer that even outperforms them in effectiveness. However, to the best of our knowledge, previous studies only take the frequency domain as a way for feature extraction or interaction, but none of them utilize it to tokenize the continuous time series for further calculation.

## 2.2 TRANSFORMER MODELS FOR SEQUENCE MODELING

Transformer layers have demonstrated powerful capabilities in handling sequential features. For text sequences, Brown et al. (2020) and Devlin et al. (2019) show strong language understanding and generation abilities based on transformer structures and pretraining tasks. In the domain of genetics, Ji et al. (2021) used three consecutive DNA bases as tokens for input and achieved excellent results after extensive training on datasets. Even in other scenarios, e.g., images (Dosovitskiy et al., 2021) and videoes (Tong et al., 2022), transformer-based models also achieve robust performance. These models first verify the fundamental units that compose data in these modalities and design appropriate pretraining tasks for these basic units, ultimately demonstrating outstanding performance.

As for the field of time series forecasting, some works exhibit similar characteristics. Nie et al. (2023) and Tonekaboni et al. (2021) treat a certain length of time point sequences as a patch (as patches in images), which is then transformed into tokens for input to the transformer layers. On the other hand, Li et al. (2023) and Yue et al. (2021) obtain basic unit representations of sequences through 1-D convolution on the sequences. However, these models select their basic units in the time domain, which often exhibits strong fluctuations, making it difficult to understand the underlying patterns of time series from a more fundamental perspective. Therefore, searching for the basic units of time series from the frequency domain perspective becomes particularly important.

## 3 FREQTST MODEL

Before introducing our FreqTST model, we formally define the time series forecasting task as follows: Given a multivariate time series sample $(\mathbf{x}_0, \ldots, \mathbf{x}_{L-1}) \in R^{M \times L}$, where L means the length of observed time steps and M means the number of features (M can be 1), we aim to predict the next T steps of the future values, which can be represented as a matrix $(\mathbf{x}_L, \ldots, \mathbf{x}_{L+T-1}) \in R^{M \times T}$.

The overview of FreqTST is illustrated in Figure 2. For each individual sequence, we employ a frequency vectorization module to convert it into a token sequence, where the Fast Fourier Transform (FFT) and embedding dictionary are utilized. Then, a transformer encoder is designed for further feature calculations. Finally, we adopt a task head for forecasting, and two designed pretraining tasks can be adopted. Data processing steps are not shown in the figure but will be introduced later.

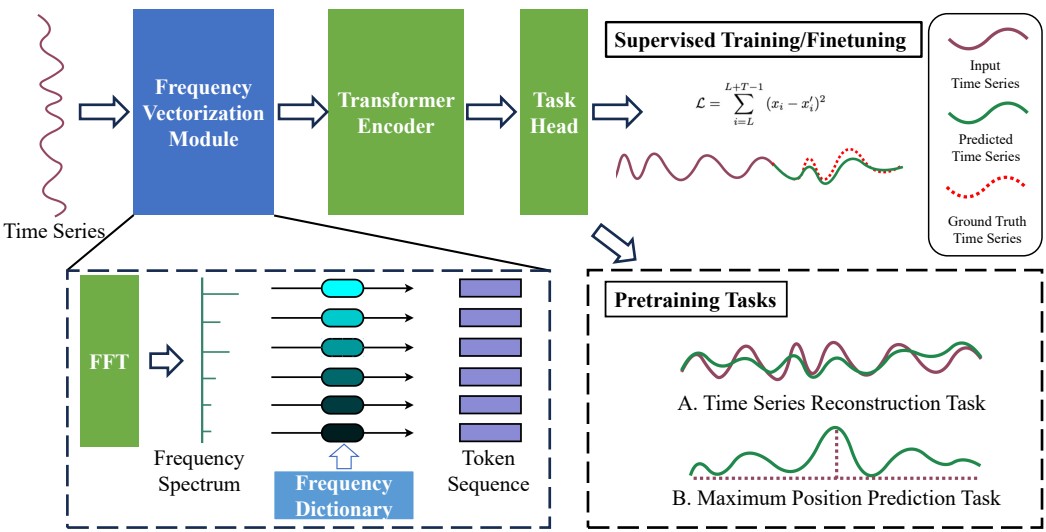

Figure 2: The overview of FreqTST model.

## 3.1 DATA PREPROCESSING

As previous studies have verified that channel-independent time series forecasting is more effective (Nie et al., 2023; Han et al., 2023), we follow this setting and divide each data sample into M sequences of length L before forecasting. So a multivariate time series $(\mathbf{x}_0, \ldots, \mathbf{x}_{L-1}) \in R^{M \times L}$ is splitted into M sequences that $\mathbf{x}^j = (x_0^{(j)}, \ldots, x_{L-1}^{(j)}) \in R^L$, where j represents the j-th sequence.

Besides, to handle data sparsity challenges in some datasets, we introduce a data augmentation method proposed by Chen et al. (2023a) to expand datasets. This method first transforms the time series into the frequency domain using FFT. Then, it randomly masks some frequencies at a certain ratio to obtain a new frequency sequence. The sequence is then transformed back into the time domain using the Inverse Fast Fourier Transform, resulting in a new time series sample that will be added to the training set. This module maintains consistency between original and augmented data and relieves the overfitting issues, which also utilizes the frequency domain as our FreqTST.

## 3.2 FREQUENCY-BASED VECTORIZATION

### 3.2.1 FROM THE TIME DOMAIN TO THE FREQUENCY DOMAIN

The procedure of vectorization is demonstrated in Figure 2. For a time series $\mathbf{x}^j \in R^L$, we apply FFT on it to form frequency sequence $(f_0^{(j)}, \ldots, f_{L-1}^{(j)}) \in C^L$. FFT is based on the Discrete Fourier Transform (DFT), which converts a discrete and finite time series into a sequence of weights of the same length. Following the formula of DFT, we can get the frequency sequence as Equation 1. Here, we omit the superscript $j$ to display the transformation clearly.

$$f_k = \sum_{n=0}^{L-1} x_n e^{-2\pi i \frac{kn}{N}}, \ k \in [0, L-1] \tag{1}$$

Then, using Euler's formula, we can obtain the Equation 2. It represents we use $L$ sinusoid and $L$ cosinuid basis wave functions and compute the similarity between the given time series and them. So $f_k$ means the projection value on frequency denoted by basis function $cos(2\pi \frac{k}{L})$ and $sin(2\pi \frac{k}{L})$.

$$f_k = \sum_{n=0}^{L-1} x_n cos(2\pi \frac{kn}{L}) - i \sum_{n=0}^{L-1} x_n sin(2\pi \frac{kn}{L}), \ k \in [0, L-1] \tag{2}$$

Furthermore, we record the similarities as coefficients $a_k$ and $b_k$, which is shown in Equation 3.

$$a_k = \sum_{n=0}^{L-1} x_n cos(2\pi \frac{kn}{L}), \; b_k = -\sum_{n=0}^{L-1} x_n sin(2\pi \frac{kn}{L}) \tag{3}$$

As a result, we can get $f_k = a_k + ib_k$, where $k \in \{0, 1, \ldots, L-1\}$, which conveys the component of the time series on the basis wave functions associated with the specific periods. In this way, we discrete the continuous time series into frequency units (tokens) that contain all information.

### 3.2.2 VECTORIZATION OF FREQUENCY UNITS

In the wake of this, we can adopt the same approach used in modeling words to generate token embedding of frequency units. In particular, we construct two fixed frequency token dictionaries $G = (\mathbf{g}_0, \mathbf{g}_1, \ldots, \mathbf{g}_{L-1}) \in R^{L \times P}$ and $H = (\mathbf{h}_0, \mathbf{h}_1, \ldots, \mathbf{h}_{L-1}) \in R^{L \times P}$, where P means the dimension of the token. Following previous studies in designing position embeddings (Vaswani et al., 2017), every element of these dictionaries is calculated by Equations 4 and 5.

$$G_{l,2k} = sin(l/10000^{2k/P}), \; G_{l,2k+1} = cos(l/10000^{2k/P}) \tag{4}$$

$$H_{l,2k} = cos(l/10000^{2k/P}), \; H_{l,2k+1} = sin(l/10000^{2k/P}) \tag{5}$$

For the k-th component of frequency units $f_k$, we can tokenize the complex scalar $a_k + ib_k$ into a real vector $\mathbf{t}_k$ using $\mathbf{g}_k$ and $\mathbf{h}_k$. The process is shown as Equation 6. The aim of introducing $W_\theta \in R^{P \times P}$ into this Equation is projecting information from the complex domain onto the real domain and co-creating the token with original information in the real domain.

$$\mathbf{t}_k = a_k \times \mathbf{g}_k + b_k \times \mathbf{h}_k \times W_\theta \tag{6}$$

After tokenizing all units, we get the final frequency spectrum sequence $T = (\mathbf{t}_0, \mathbf{t}_1, \ldots, \mathbf{t}_{L-1}) \in R^{P \times L}$, where each vector can be fed into transformer naturally as a semantic token.

In this way, we build a shared dictionary applicable to any time series and map each time series into the frequency spectrum sequence with discrete frequencies as the basic unit. Considering the calculation process of FFT, this sequence fully integrates the information of all value points on the time series so that the transformation is unscathed. Furthermore, it is worth mentioning that we only retain the first $L' = (L//2) + 1$ frequency of FFT in further calculations, due to the conjugate symmetry after the transformation. This not only preserves full information but also improves computational efficiency (Cooley & Tukey, 1965).

### 3.3 TRANSFORMER ENCODER

As shown in 2, the frequency spectrum sequence will be adopted for further computation using a vanilla Transformer Encoder. The attention mechanism in the encoder allows the information between different frequencies to be integrated, leveraging long-range dependencies between different frequency tokens for the original time series. Finally, we will obtain a final comprehensive representation $Z = (\mathbf{z}_0, \mathbf{z}_1, \ldots, \mathbf{z}_{L'-1}) \in R^{P \times L'}$ that represents the time series.

### 3.4 PRETRAINING TASKS

As the tokenization is conducted on the frequency domain, we design two specific pretraining tasks to align the time and frequency domain. Here, we do not simply follow the Masked Language Model (MLM) pretraining task in Devlin et al. (2019), and there are two reasons: 1) it makes the model believe there are strong relations between the units of frequency spectrum sequence while they are independent with each other in fact. 2) the downstream forecasting task is strongly associated with time domain information, so only pretraining frequency tokens limit the forecasting performance.

**Time Series Reconstruction (TSR task).** This task aims to reconstruct time-domain information from frequency, which aligns the time and frequency domain information. Given a time series $X = (x_0, \ldots, x_{L-1}) \in R^L$, the objective of this pretraining task is to reconstruct the original time series based on its final representations $Z = (\mathbf{z}_0, \mathbf{z}_1, \ldots, \mathbf{z}_{L'-1}) \in R^{P \times L'}$. We add an MLP

Table 1: Statistics of adopted datasets.

| Datasets | Weather | Electricity | Ettm2 | Etth2 |
|---|---|---|---|---|
| #Feature | 21 | 321 | 7 | 7 |
| Length | 52,140 | 26,305 | 69,680 | 17,420 |
| Data size | 6.28M | 91.10M | 9.22M | 2.30M |

task head after Transformer Encoder to calculate $X' = (x'_0, \ldots, x'_{L-1}) \in R^L$, which flattens the final representation $Z$ into a vector of dimension $R^{PL'}$ and then maps the vector to predict the reconstructed time series $X'$ using an MLP layer. The loss function of this task is $\mathcal{L}_1 = \sum_{i=0}^{L-1} (x_i - x'_i)^2$.

**Maximum Position Prediction (MPP task).** This pretraining task targets to focus on learning the patterns of temporal changes and sequence peaks, addressing the limitation of frequency domain representation being insensitive to amplitude variations. This task also generates a sequence $X' = (x'_0, \ldots, x'_{L-1}) \in R^L$ based on $Z \in R^{P \times L'}$ to predict the maximum position of the original time series. We use another task head to flatten and map the final representation $Z$, and adopt Softmax operation at the final step. Consequently, all values in $X'$ are in the range of 0 and 1, which means the probability of the current position being the maximum value. Similarly, assuming the ground truth is represented as a one-hot vector $\mathbf{w}$, the loss function of this task is $\mathcal{L}_2 = \sum_{i=0}^{L-1} (w_i - x'_i)^2$.

The two pretraining tasks are utilized in our pretraining experiments, and the final loss $L$ calculation is given by Equation 7, where $\beta$ denotes the weight value and is a hyper-parameter.

$$\mathcal{L}_{pretraining} = \mathcal{L}_1 + \beta \times \mathcal{L}_2 \tag{7}$$

## 3.5 MODEL TRAINING AND TRANSFERRING

**Fine-tuning/Supervised Training.** The process of supervised learning and fine-tuning is essentially the same. After obtaining the final representation $Z \in R^{P \times L'}$, we use an MLP layer to generate future time steps $X' = (x'_L, \ldots, x'_{L+T-1}) \in R^T$. Given the groundtruth $X = (x_L, \ldots, x_{L+T-1}) \in R^T$, the loss function is $\mathcal{L} = \sum_{i=L}^{L+T-1} (x_i - x'_i)^2$. The difference between fine-tuning and supervised learning is that we load parameters from a pretrained model for fine-tuning and initialize parameters randomly for supervised learning.

**Transfer Learning on Zero-shot Experiment.** As the trained frequency dictionaries can be utilized across various datasets, our model has strong zero-shot transfer learning abilities. To be more specific, FreqTST can be trained on a large-scale dataset and tested on unseen datasets. Further experiments are conducted in Section 4.3.2 to verify its effectiveness.

## 4 EXPERIMENTS

### 4.1 EXPERIMENTAL SETTINGS

**Datasets**. We use the four widely used time series datasets and follow previous settings for evaluation(Wu et al., 2021): 1) Weather dataset that describes meteorological information of a German town with 21 features like rainfall and wind speed. 2) Electricity dataset that contains hourly electricity consumption of 321 customers. 3) ETTm2 and 4) ETTh2 gathered from electricity transformers with different resolutions (minute and hour) with 7 features. Some statistics are shown in Table 1.

**Metrics**. Following previous studies (Nie et al., 2023; Zhou et al., 2022b), we use MSE and MAE to evaluate the model performance. Given the prediction $\hat{Y} = (\hat{y}_1, \hat{y}_2, \ldots, \hat{y}_T)$ and ground truth $Y = (y_1, y_2, \ldots, y_T)$, MSE and MAE are calculated by Equation 8, the smaller the better.

$$MSE(\hat{Y}, Y) = \frac{1}{T} \sum_{i=1}^{T} (\hat{y}_i - y_i)^2, \quad MAE(\hat{Y}, Y) = \frac{1}{T} \sum_{i=1}^{T} |\hat{y}_i - y_i| \tag{8}$$

**Baselines**. We utilize well-known frequency-based models time series forecasting models, such as Autofomer (Wu et al., 2021), FEDformer (Zhou et al., 2022b), and FiLM (Zhou et al., 2022a), as baseline models. Besides, Dlinear (Zeng et al., 2023) is added to exhibit the long-term prediction effectiveness of our model. State-of-the-art transformer-based methods, including Informer (Zhou et al., 2021) and PatchTST (Nie et al., 2023), are also used.

**Implementation Details**. For fair comparisons, we fix the look-back window L into 96 and set the forecasting length $T \in \{96, 192, 336, 720, 960, 1120, 1280, 1440\}$, where the last four settings are chosen for long-term predictions. Polit experiments show that pretraining and end-to-end fine-tuning in 40 and 100 epochs, respectively, are most suitable here. For frequency spectrum representations, the embedding dimension P is set to 16, and the representation dimension of transformers is kept as 128. The loss weight $\beta$ is altered in 0.25, 0.5, 0.75, 1, 2, 3 to balance the bias of two pretraining tasks. Furthermore, we set the transformer layer to 3 and batch size to 64. Experiments are running on 4*GeForce RTX 3090 GPUs. Note that the model with a pretraining tag means this model adopts the dataset for model pretraining before fine-tuning, but no extra data is adopted.

## 4.2 MAIN EXPERIMENTS

Table 2: Main experimental results on various datasets with forecasting length in 96, 192, 336, 720. The best results among methods are in boldface, and the second best results are underlined.

| Models | Metric | Weather | | | | Electricity | | | | ETTh2 | | | |
|---|---|---|---|---|---|---|---|---|---|---|---|---|---|
| | | 96 | 192 | 336 | 720 | 96 | 192 | 336 | 720 | 96 | 192 | 336 | 720 |
| Informer | MSE | 0.356 | 0.525 | 0.591 | 1.013 | 0.308 | 0.342 | 0.338 | 0.351 | 0.341 | 0.606 | 0.642 | 0.750 |
| | MAE | 0.419 | 0.506 | 0.541 | 0.739 | 0.396 | 0.428 | 0.422 | 0.429 | 0.404 | 0.515 | 0.556 | 0.608 |
| Autoformer | MSE | 0.323 | 0.336 | 0.360 | 0.416 | 0.237 | 0.263 | 0.275 | 0.369 | 0.219 | 0.247 | 0.274 | 0.326 |
| | MAE | 0.362 | 0.375 | 0.387 | 0.424 | 0.348 | 0.369 | 0.377 | 0.438 | 0.325 | 0.348 | 0.371 | 0.409 |
| FEDformer | MSE | 0.254 | 0.286 | 0.343 | 0.415 | 0.188 | 0.197 | 0.214 | 0.244 | 0.213 | 0.249 | 0.265 | 0.324 |
| | MAE | 0.340 | 0.349 | 0.382 | 0.426 | 0.303 | 0.311 | 0.328 | 0.353 | 0.318 | 0.348 | 0.361 | 0.411 |
| FiLM | MSE | 0.245 | 0.300 | 0.346 | 0.420 | 0.204 | 0.215 | 0.230 | 0.306 | 0.222 | 0.244 | 0.271 | 0.331 |
| | MAE | 0.317 | 0.358 | 0.385 | 0.427 | 0.313 | 0.322 | 0.338 | 0.401 | 0.330 | 0.340 | 0.367 | 0.414 |
| Dlinear | MSE | 0.195 | 0.236 | 0.283 | 0.352 | 0.195 | 0.193 | 0.206 | 0.242 | 0.190 | 0.228 | 0.250 | 0.308 |
| | MAE | 0.253 | 0.294 | 0.333 | 0.391 | 0.277 | 0.280 | 0.295 | 0.329 | 0.300 | 0.333 | 0.354 | 0.403 |
| PatchTST | MSE | 0.178 | 0.225 | 0.277 | 0.351 | 0.186 | 0.189 | 0.205 | 0.246 | 0.182 | 0.225 | 0.256 | 0.324 |
| | MAE | 0.219 | 0.259 | 0.297 | 0.346 | 0.268 | 0.273 | 0.288 | 0.321 | 0.289 | 0.322 | 0.346 | 0.394 |
| PatchTST$_{pretraining}$ | MSE | 0.172 | 0.217 | 0.274 | 0.350 | 0.184 | 0.189 | 0.204 | 0.246 | **0.178** | **0.218** | 0.247 | 0.320 |
| | MAE | **0.214** | **0.255** | **0.296** | **0.345** | 0.267 | 0.272 | 0.288 | 0.321 | **0.284** | **0.317** | 0.340 | 0.394 |
| FreqTST | MSE | **0.169** | **0.210** | **0.258** | **0.326** | **0.179** | **0.183** | **0.197** | **0.235** | 0.185 | 0.225 | **0.236** | **0.291** |
| | MAE | 0.229 | 0.267 | 0.307 | 0.358 | **0.266** | **0.271** | **0.287** | **0.320** | 0.294 | 0.327 | **0.338** | **0.381** |

Experimental results on three datasets are shown in Table 2. Note the performance of FreqTST is based on supervised learning without pretraining. From the table, we have the following observations: Firstly, our FreqTST outperforms all other supervised baseline models, including PatchTST and other frequency-based models, which verifies the effectiveness of our model. Secondly, compared with PatchTST$_{pretraining}$ that contains an extra pretraining step, FreqTST achieves better or comparable performances with lower computation costs (without pretraining). FreqTST shows superior results in the large dataset (Electricity) and in long-term forecasting (ETTh2), and indicates stronger stability than PatchTST$_{pretraining}$ for its better MSE performances in the Weather dataset. To summarize, these results demonstrate the usefulness of our frequency-based vectorization strategy for time series forecasting in various datasets than SOTA models.

Going a step further, we carry out the pretraining and fine-tuning strategy to show the effects of the proposed two pretraining tasks. Four models are utilized in this experiment with ablations of the two tasks: 1) FreqTST, 2) FreqTST$_{OnlyTSR}$ that uses the TSR task, 3) FreqTST$_{OnlyMPP}$ that adopts the MPP task, and 4) FreqTST$_{pretraining}$ using two tasks.

As illustrated in Figure 3, FreqTST$_{pretraining}$ method outperforms the supervised version in most cases, which indicates the usefulness of the pretraining strategy. Besides, results in this figure also indicate that two designed pretraining tasks can bring improvements simultaneously, as there are obviously drops when only one pretraining task is executed.

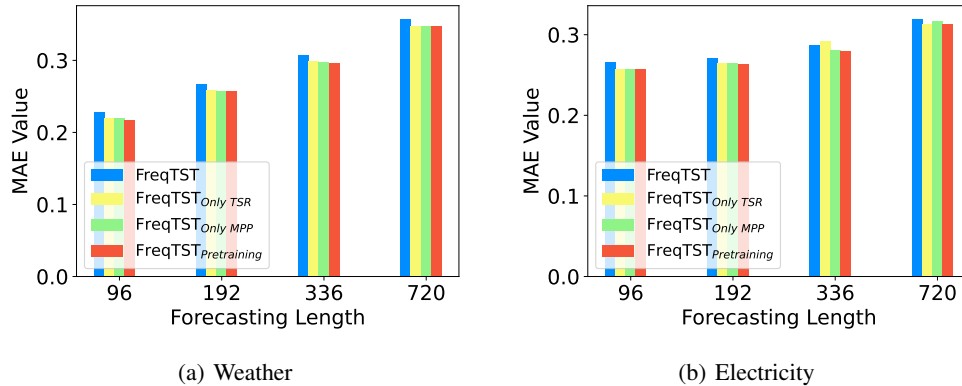

(a) Weather  (b) Electricity

Figure 3: Ablation experimental results of pretraining tasks.

Table 3: Long-term forecasting results. '-' means the model cannot finish this task on our server.

| Models | Metric | Weather | | | | Electricity | | | | Etth2 | | | |
|---|---|---|---|---|---|---|---|---|---|---|---|---|---|
| | | 960 | 1120 | 1280 | 1440 | 960 | 1120 | 1280 | 1440 | 960 | 1120 | 1280 | 1440 |
| Autoformer | MSE | 0.438 | 0.435 | 0.451 | 0.463 | 0.354 | 0.388 | 0.401 | 0.388 | 0.367 | 0.390 | 0.420 | 0.460 |
| | MAE | 0.437 | 0.421 | 0.434 | 0.445 | 0.431 | 0.452 | 0.459 | 0.453 | 0.442 | 0.459 | 0.480 | 0.513 |
| FEDformer | MSE | 0.421 | 0.464 | 0.454 | 0.450 | 0.270 | 0.288 | 0.277 | 0.336 | 0.345 | 0.417 | 0.432 | 0.475 |
| | MAE | 0.414 | 0.454 | 0.447 | 0.440 | 0.369 | 0.384 | 0.379 | 0.421 | 0.431 | 0.482 | 0.494 | 0.515 |
| DLinear | MSE | 0.365 | 0.372 | 0.378 | 0.386 | 0.260 | **0.274** | **0.287** | 0.299 | 0.353 | 0.388 | 0.425 | 0.476 |
| | MAE | 0.399 | 0.406 | 0.411 | 0.416 | 0.344 | 0.356 | 0.367 | 0.375 | 0.434 | 0.457 | 0.479 | 0.509 |
| PatchTST$_{pretraining}$ | MSE | 0.376 | 0.388 | 0.396 | 0.410 | 0.280 | - | - | - | 0.366 | 0.392 | 0.444 | 0.464 |
| | MAE | **0.366** | **0.372** | **0.378** | **0.385** | 0.354 | - | - | - | 0.426 | 0.441 | 0.476 | 0.484 |
| FreqTST | MSE | **0.351** | **0.357** | **0.360** | **0.370** | **0.253** | 0.275 | 0.289 | **0.294** | **0.299** | **0.342** | **0.374** | **0.382** |
| | MAE | 0.378 | 0.383 | 0.390 | 0.398 | **0.335** | **0.355** | **0.365** | **0.366** | **0.386** | **0.419** | **0.447** | **0.455** |

## 4.3 ANALYSES

### 4.3.1 LONG-TERM FORECASTING

To verify the forecasting performance of FreqTST in the long-term, we propose to set the forecasting length to $\{960, 1120, 1280, 1440\}$. The results are summarized in Table 3. Firstly, FreqTST beats all baseline models in most metrics, except the MAE performances in the Weather dataset, showing that our model is good at long-term prediction. Secondly, PatchTST with pretraining, the strongest baseline in Table 2, performs even worse than DLinear in some datasets, indicating this task is challenging. Thirdly, PatchTST cannot finish the forecasting, showing that its structure is not efficient enough. These results demonstrate the effectiveness and efficiency of frequency spectrum representation in FreqTST, which has strong capabilities to generate context with a long range.

### 4.3.2 ZERO-SHOT TRANSFER LEARNING

One of the most powerful abilities of transformers with dictionaries is their generalization feature. As we propose to model time series with frequency tokens, we are going to verify the zero-shot transfer learning ability of our FreqTST model. To be specific, we pretrain the FreqTST and PatchTST model on Electricity (a large dataset) and test the model performance on Etth2 (a small dataset) without training, which is a zero-shot setting. Besides, we conduct the same evaluation strategy on two supervised models Autoformer and FEDformer for comparison.

Experimental results are demonstrated in Figure 4. Both FreqTST and FreqTST$_{pretraining}$ show more promising performances on the zero-shot setting than PatchTST, which even outperform some supervised models (AutoFormer and FEDFormer). These results indicate our model has capable transfer ability due to the unified frequency units and common dictionary among all kinds of time

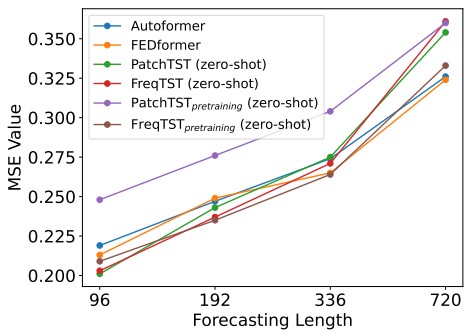 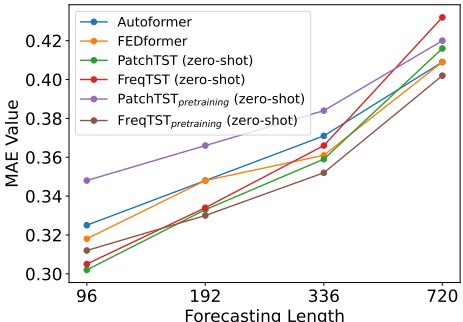

Figure 4: Forecasting results on zero-shot settings for the Etth2 dataset.

Table 4: Comparisons with data augmentation enhanced frequency-based baselines.

| Models | | Metric | Weather | | | | Electricity | | | | Etth2 | | | |
|---|---|---|---|---|---|---|---|---|---|---|---|---|---|---|
| | | | 96 | 192 | 336 | 720 | 96 | 192 | 336 | 720 | 96 | 192 | 336 | 720 |
| Data Augmentation | Informer | MSE | 0.378 | 0.461 | 0.479 | 0.731 | 0.286 | 0.302 | 0.306 | 0.318 | 0.347 | 0.574 | 0.665 | 0.726 |
| | | MAE | 0.425 | 0.462 | 0.486 | 0.630 | 0.379 | 0.396 | 0.399 | 0.406 | 0.403 | 0.498 | 0.563 | 0.596 |
| | FiLM | MSE | 0.244 | 0.309 | 0.365 | 0.424 | 0.185 | 0.206 | 0.226 | 0.342 | 0.231 | 0.244 | 0.267 | 0.329 |
| | | MAE | 0.316 | 0.361 | 0.404 | 0.433 | 0.295 | 0.311 | 0.332 | 0.428 | 0.336 | 0.341 | 0.362 | 0.413 |
| | Autoformer | MSE | 0.273 | 0.300 | 0.365 | 0.432 | 0.177 | 0.194 | 0.228 | 0.266 | 0.237 | 0.256 | 0.276 | 0.374 |
| | | MAE | 0.339 | 0.359 | 0.399 | 0.439 | 0.292 | 0.305 | 0.336 | 0.364 | 0.331 | 0.356 | 0.379 | 0.435 |
| | FEDformer | MSE | 0.227 | 0.297 | 0.338 | 0.395 | **0.178** | 0.188 | 0.209 | **0.225** | 0.208 | 0.260 | 0.274 | 0.320 |
| | | MAE | 0.308 | 0.357 | 0.382 | 0.407 | 0.292 | 0.300 | 0.322 | 0.336 | 0.313 | 0.357 | 0.368 | 0.404 |
| FreqTST | | MSE | **0.169** | **0.210** | **0.258** | **0.326** | 0.179 | **0.183** | **0.197** | 0.235 | **0.185** | **0.225** | **0.236** | **0.291** |
| | | MAE | **0.229** | **0.267** | **0.307** | **0.358** | 0.266 | **0.271** | **0.287** | 0.320 | **0.294** | **0.327** | **0.338** | **0.381** |

series. Besides, FreqTST with pretraining is superior to FreqTST, indicating the pretraining strategy contributes significantly to our model's capacity for generalization.

### 4.3.3 COMPARISON TO FREQUENCY-BASED MODEL

To verify whether the improvements of FreqTST are caused by the proposed frequency vectorization strategy or data augmentations, we compare our model with other frequency-based models that are enhanced with data augmentations in Section 3.1. Results are exhibited in Table 4. We find that most frequency-based baseline models perform even worse than those in Table 2, which means data augmentations do not provide improvements for them. Besides, our Frequency outperforms all baselines with an average 9.72% drop on MSE and 8.07% drop on MAE, indicating that our frequency vectorization module plays an essential role in achieving precision time series forecasting.

## 5 CONCLUSIONS

In this study, we propose an effective method, named FreqTST, to model time series as text sequences by utilizing the frequency and time domain characters of time series. Different from prior works that only extract features from the frequency domain, we discrete the time series into frequency units, tokenize them using a common dictionary and design two novel pretraining tasks. The architecture makes full use of the modeling capacity of transformers for discrete and semantic token sequence, and outperforms other supervised models by a large margin. Besides, it also shows excellent ability for long-term forecasting and zero-shot transfer learning.

Our model also shows promising potential for future work. With the unified discretion of time series and a common dictionary for frequency units, it could be pretrained on all kinds of time series and generalized to unseen datasets for various downstream tasks.

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
