# OpenReview forum: "Modeling Time Series as Text Sequence A Frequency-vectorization Transformer for Time Series Forecasting"
_ICLR.cc/2024/Conference — Submitted to ICLR 2024_

### Official Review · Reviewer_EgZL · 2023-10-24

**Soundness:** 2 fair
**Presentation:** 3 good
**Contribution:** 3 good
**Rating:** 5
**Confidence:** 4

**Summary:**

This paper introduces FreqTST, a Transformer-based model for time series forecasting. Unlike prior approaches that tokenize time series in the time domain, FreqTST employs the frequency spectrum as a common dictionary for tokenization. It incorporates two pretraining tasks: time series reconstruction and maximum position prediction, aimed at aligning the frequency and time domains. The experimental results highlight FreqTST's superiority over existing state-of-the-art models. The model demonstrates a certain level of transferability across different datasets.

**Strengths:**

- This paper is well-structured and easy to follow.
- The concept of utilizing the frequency spectrum as a tokenization dictionary is novel and intriguing.
- The transferability across different datasets is a bonus.

**Weaknesses:**

- Becauses the frequencies of the DFT basis dependences on sequence length, my main concern is that the tokenization process is highly dependent on the length of the input sequence $L$:
  1. The size of token dictionaries, $G, H \in \mathbb{R}^{L\times P}$ vary with the input sequence length $L$.
  2. The correspondence between keys in the dictionary and their frequencies also changes with varying sequence lengths: when embedding a sequence of length 10, $g_5, k_5$ in the fixed dictionary correspond to frequency $\frac{1}{2}$, i.e. period of $2$. While for sequence of length 20, they correspond to frequency $\frac{1}{4}$, i.e. period of $4$.

  Therefore, the tokenization highly dependences on the seqeunce length and we need a specific dictionary for each different length, even on the same data set. This limits the generalization of the proposed model.

- The tokenization should be orthogonal to pretraining task. However, it seems that FreqTST highly relies on pretraining to achieve SOTA performance.

- The paper uses a fixed input length of 96 in the main experiments, while the forecasting lengths are substantially larger (e.g., 336, 720). This design choice may not be realistic. Moreover, PatchTST and DLinear can get better forecasting accuracy with longer look-back window.

- The claim of zero-shot transferability might be overstated. In section 4.3.2, the model is trained on Electricity and tested on ETTh2 to show its transferability. However, both these two datasets are hourly-sampled and have a period of 24 hours. I infer that this pre-trained model is unlikely to achieve satisfactory results on weather as it is sampled every 10 minutes. Thus "These results indicate our model has capable transfer ability due to the unified frequency units and common dictionary among all kinds of time series" seems to be overclaimed.

**Questions:**

1. Section 4.1 mentions experiments conducted on the ETTm2 dataset, but the results are missing. Can you provide the outcomes for this dataset?
2. Why MSE was chosen instead of cross-entropy in Maximum Position Prediction task, consdering this is a classification task and the target is a one-hot vector?
3. Please compare FreqTST with DLinear and PatchTST with varying look-back window length $L$.
4. Please evaluate the computation efficiency (memory occupation and speed) of FreqTST with different look-back window $L$ and number of features $M$.

---

> ### Author Response · Authors · 2023-11-23
>
> Thanks for your recognition and kindly comments.
>
> Q1: Dependency on sequence length. A very insightful observation. Due to the limitations of DFT, we acknowledge that our tokenization method is highly correlated with input length. However, this does not imply weak generalization of our method. On the one hand, upon careful examination of the expressions for G and H, we can find that regardless of changes in input length, as long as the embedding dimensions remain constant, the basis vectors at each position in G and H will not change. This ensures the generalization of token dictionaries. On the other hand, any time series can be partitioned into fixed-length segments. For instance, a time series of length 200 can be divided into 105 subsequences of length 96 (105 = 200 - 96 + 1), and these subsequences can be sampled at regular intervals for prediction. The average results from these predictions can be considered as the final prediction result. This approach allows for extension to time series tasks of arbitrary lengths.
>
> Q2: Relation between tokenization and pretraining task. There is probably some misunderstanding here. Firstly, both the TSR and MPP pretraining tasks can be trained using the time series itself and the position of the maximum value, independent of specific tokenization. Secondly, in the main experiment, FreqTST achieves state-of-the-art results without pretraining tasks.
>
> Q3: Fixed input length. We fixed the input length to 96 because we aimed to model scenarios with missing data, such as in medical contexts, which are common occurrences.
>
> Q4: Completion of zero-shot experiments. It is very kind of you to point out this problem. In fact, we conduct more experiments but only represent one sample in our paper. Here are more results that show the powerful transferibility of FreqTST.
> |        Models       | Dataset | Weather |       |       |       | Ettm2 |       |       |       | Etth2 |       |       |       |
> |:-------------------:|:-------:|:-------:|:-----:|:-----:|:-----:|:-----:|:-----:|:-----:|:-----:|:-----:|:-----:|:-----:|:-----:|
> |                     |  Metric |    96   |  192  |  336  |  720  |   96  |  192  |  336  |  720  |   96  |  192  |  336  |  720  |
> |  PatchTST_ sup_zero |   MSE   | 0.232   | 0.276 | 0.338 | 0.418 | 0.173 | 0.193 | 0.236 | 0.286 | 0.201 | 0.243 | 0.275 | 0.354 |
> |                     |   MAE   | 0.273   | 0.306 | 0.350 | 0.402 | 0.294 | 0.309 | 0.342 | 0.373 | 0.302 | 0.333 | 0.359 | 0.416 |
> | PatchTST_ self_zero |   MSE   | 0.234   | 0.280 | 0.329 | 0.397 | 0.170 | 0.190 | 0.220 | 0.275 | 0.248 | 0.276 | 0.304 | 0.360 |
> |                     |   MAE   | 0.281   | 0.314 | 0.344 | 0.385 | 0.290 | 0.304 | 0.325 | 0.361 | 0.348 | 0.366 | 0.384 | 0.420 |
> |  FreqTST_ sup_zero  |   MSE   | 0.204   | 0.249 | 0.304 | 0.392 | 0.177 | 0.206 | 0.276 | 0.405 | 0.203 | 0.237 | 0.271 | 0.361 |
> |                     |   MAE   | 0.258   | 0.293 | 0.334 | 0.396 | 0.297 | 0.322 | 0.375 | 0.455 | 0.305 | 0.334 | 0.366 | 0.432 |
> |  FreqTST_ self_zero |   MSE   | 0.247   | 0.264 | 0.328 | 0.373 | 0.142 | 0.183 | 0.205 | 0.258 | 0.209 | 0.235 | 0.264 | 0.333 |
> |                     |   MAE   | 0.271   | 0.290 | 0.335 | 0.362 | 0.257 | 0.291 | 0.310 | 0.346 | 0.312 | 0.330 | 0.352 | 0.402 |
>
> Q5: Missing ETTm2 dataset. We use the ETTm2 dataset to evaluate zero-shot capacity of FreqTST, which is demonstrated by the table in Q4. We also evaluate main models on ETTm2, and the results are below.
> |      Models     | Metric |   96  |  192  |  336  |  720  |
> |:---------------:|:------:|:-----:|:-----:|:-----:|:-----:|
> |  FreqTST_ self  |   MSE  | 0.119 | 0.150 | 0.190 | 0.245 |
> |                 |   MAE  | 0.231 | 0.262 | 0.293 | 0.333 |
> |   FreqTST_ sup  |   MSE  | 0.119 | 0.154 | 0.190 | 0.243 |
> |                 |   MAE  | 0.233 | 0.269 | 0.301 | 0.340 |
> | PatchTST_ self_ |   MSE  | 0.120 | 0.147 | 0.180 | 0.238 |
> |                 |   MAE  | 0.231 | 0.260 | 0.286 | 0.331 |
>
> Q6: Cross entropy for MPP task. Thanks for your valuable suggestion. We conducted validation experiments using the cross entropy loss and found that the results were not as good as those using MSE. We think this is because the pre-training loss function is the sum of the loss functions from the two pre-training tasks. If the loss functions of the two pre-training tasks are different, the optimization space of the entire loss function will be more rugged, making it difficult to converge to the global optimal solution.
>
> Q7: Results with varying length L. As is mentioned in Q3, we focus on the scene that specializes in short input, so we do not include this experiment due to the limitation of time and compute resources. We will do our best to do this experiment in the near future.

---

### Official Review · Reviewer_ggum · 2023-10-26

**Soundness:** 2 fair
**Presentation:** 3 good
**Contribution:** 3 good
**Rating:** 6
**Confidence:** 5

**Summary:**

This paper introduces a frequency-based time series tokenization approach, diverging from the conventionally employed patching strategy. The proposed methodology is somewhat novel, simple, and effective. I particularly appreciate the insights shared in Sec 3.2 and recognize the potential this method offers for pioneering time series pre-training across domains. In terms of experiments, the presented results offer a fairly comprehensive validation of the method's effectiveness. Nevertheless, certain nuances in the paper's composition could benefit from refinement. Please refer to my comments below for specifics.

**Strengths:**

1. The motivation behind this research is compelling, setting a strong foundation for future studies on time series pre-training across multiple domains.
2. The concept of tokenizing time series based on frequency units offers a fresh perspective.
3. The experiments conducted provide a thorough assessment, effectively showcasing the efficacy of the proposed method.

**Weaknesses:**

1. The technical soundness of Sec 3 should be further enhanced. See my listed comments & questions below.
2. The presentation needs to be further improved, and some claims fail to be justified with evidence or in-depth discussion.

**Questions:**

I have the following detailed questions and comments:

- The claim in the third paragraph of the introduction that "... transformer architectures cannot be directly applied to model time series data due to the lack of a clear tokenization scheme" seems unsubstantiated. Recent studies, such as [1], have showcased the feasibility of such an application.
- In Fig 1, the demarcation between pre-training and downstream adaptation is not clearly delineated. This aspect remains vague in Sec 3, and one has to refer to the discourse in Sec 4.2 for clarity. For instance, the statement "the difference between fine-tuning and supervised learning is that we load ..." in Sec 3.5 appears ambiguous. Furthermore, the utilization of L_pre and L in Sec 3 isn't elaborated upon.
- The rationale for introducing G and H in Sec 3.2.2 is unclear. The link between positional encoding (specifically, Eq. 4 and 5) and the formulation of G and H requires more substantial justification.
- Eq. 6 presents a transformation of f_k. Based on my interpretation, the core concept hinges on retaining a set of shared bases, namely G and H. Nevertheless, a deeper analysis or discourse is essential for grasping the underlying principles. For instance, elucidation on the relationship between t_k and f_k and the justification for selecting Eq.6 as an optimal solution would be enlightening.
- Conducting evaluations on a broader set of datasets, for instance, the remaining three ETT datasets and M4, would enhance the robustness of this work.

[1] Gruver, N., Finzi, M., Qiu, S., & Wilson, A. G. (2023). Large Language Models Are Zero-Shot Time Series Forecasters. arXiv preprint arXiv:2310.07820.

---

> ### Author Response · Authors · 2023-11-23
>
> We thank the reviewer a lot for raising this comment and have modified the corresponding presentations to eliminate the confusion in our paper.
>
> Q1: Overclaimed. Thank you for the reminder. After carefully reading the paper you recommended (sorry we did not notice this study previously), we found that the claims in our paper are not proper, and we will revise  by "... effectively applied to model time time series data". At the same time, we would like to emphasize as the counting ability of LLM is poor and the experimental results are quite mediocre, this work does not impact the value of our work.
>
> Q2: Vagueness between pre-training and downstream adaption. There is probably some misunderstanding here. In Fig 1, we did not intend to introduce pretraining tasks and downstream tasks; rather, we aimed to showcase the motivation behind our model design and its core modeling ideas. However, there is a clear boundary between pretraining tasks and downstream tasks in Fig 2. Additionally, in the first paragraph of Sec 3, we provided a detailed definition of downstream tasks, complementing the discussion of pretraining tasks in Sec 3.4. The confusion may arise from the similarity between fine-tuning and supervised learning, as we conducted fine-tuning directly on downstream tasks. Therefore, the processes of fine-tuning and supervised learning are identical, with the distinction being that fine-tuning is the next stage after pretraining, while supervised learning is a complete training process. Besides, L is defined in the first paragraph of Sec 3 when L_pre was not used in Sec 3.
>
> Q3: Motivation for frequency token formulation. The design principles for G and H align with the design principles of position encoding in Transformer. Firstly, each frequency position needs a unique and deterministic representation. Secondly, the representation distance between any two positions should be consistent across time series. Lastly, this representation should easily generalize to longer sequences with bounded values, as time series can be very long. Based on these principles, we constructed two frequency vector spaces, where G and H represent the basis vector of real and imaginary domains, respectively. After applying FFT to a time series, the coefficients obtained in the real and imaginary domains serve as coordinates for the basis of the two vector spaces. Finally, we use a linear mapping matrix to transform vectors from the imaginary domain to the real domain, and then sum the two, resulting in the final vector in the frequency space. This allows us to represent a time series as vectors in a frequency vector space.
>
> Q4: Limited Dataset. To address your concern, we have tried our best to add the required results. The table below presents the experimental results of our model on the Ettm2 dataset.
> |      Models     | Metric |   96  |  192  |  336  |  720  |
> |:---------------:|:------:|:-----:|:-----:|:-----:|:-----:|
> |  FreqTST_ self  |   MSE  | 0.119 | 0.150 | 0.190 | 0.245 |
> |                 |   MAE  | 0.231 | 0.262 | 0.293 | 0.333 |
> |   FreqTST_ sup  |   MSE  | 0.119 | 0.154 | 0.190 | 0.243 |
> |                 |   MAE  | 0.233 | 0.269 | 0.301 | 0.340 |
> | PatchTST_ self_ |   MSE  | 0.120 | 0.147 | 0.180 | 0.238 |
> |                 |   MAE  | 0.231 | 0.260 | 0.286 | 0.331 |

---

### Official Review · Reviewer_PGhs · 2023-10-30

**Soundness:** 1 poor
**Presentation:** 1 poor
**Contribution:** 2 fair
**Rating:** 3
**Confidence:** 4

**Summary:**

This work tokenizes continuous valued time-series data using the discrete nature of the temporal sampling. Here tokens correspond to the frequency basis vector of the Fourier transform. This differs from some previous works which have a continous embedding as input into a transformer. This work produced results that are comparable and in some metrics better than the previous state of the art on a subset of temporal benchmarks.

**Strengths:**

The authors addressed a primary difficulty in applying transformers to continuous-valued time series: transformer models were designed to predict discrete tokens and time series data values are continuous. Bridging this discontinuity has been the subject of many works. The approach here is clever in that discretization naturally arises from the discrete Fourier transform. By leveraging this discretization in time, the authors were able to produce discrete tokens while maintaining a continuous representation of the time series values.

Although this is not the only work that uses the Fourier domain, this Fourier representation naturally captures temporal dynamics and long-time behaviors. I believe this representation is more information-rich than the temporal domain. The Fourier domain allows the network to more easily remove noise and less relevant dynamics by masking a single (or a neighborhood) of coefficients.

**Weaknesses:**

There are a few general topics in which this paper needs significant improvements. Below I describe them starting from the least and moving to the most severe.

# Language
The writing of the paper is overall quite poor with respect to the use of English and Latex, the clarity, and the way information is and is not provided. If the authors are not native English speakers I recommend asking a native English speaker to provide more in-depth feedback. I have also found that software like grammerly is very helpful.

## Writing
1. The author uses words that do not exist in English (cosinuid should be cosine), and employs similar-sounding words that have completely different meanings (discretion is to speak in a way to not annoy or cause offense).
2. This paper's grammar needs improvements, below are two such examples.
    * we discrete the time series into frequency units
    * It represents we use L sinusoid
3. The biggest issue with language is that this work is extremely vague which makes the arguments very weak and unconvincing. This vagueness at times leaves me to wonder how well the authors understand what they are saying. This is unfortunate because the authors may be experts with intimate knowledge of this subject, but their language does not always portray this. By detailing their ideas with more precise language the authors can write a significantly more compelling study that will also demonstrate the authors' expertise in the field. Below are a few of many examples.
    * Transformer-based models achieve better performances in long-term time series forecasting than previous models, *so more and more transformer-based models are designed for better performance*
        * What about transformer models achieve better performance and why do people want to use them? Saying that "more and more" models are being made adds no information and sounds too colloquial. Either list some recent advancements or remove this portion.
    * Another research trend is the utilization of the frequency domain in forecasting, as time series in the time domain can be transformed into the frequency domain. An intuitive way is to use the frequency domain to extract more features.
        * "time series in the time domain" is redundant, time series are in the time domain. A decomposition into the Fourier or any other domain has a different name. What do you mean by "extract more features"? Information is either conserved or lost in Fourier transformations, this process cannot create more information or features. Instead, the Fourier transform has the same number of degrees of freedom by representing the data differently. Consequently, this transformation can elucidate potential patterns or information content of interest. Your vague statement leaves me wondering what you are interested in and if you think that the Fourier transform is creating new information.
4. What are these data augmentation methods mentioned in section 4.3.3?

## Latex
1. Only mathematical variables are supposed to be italicized, not function names. Also, please fix parentheses so they are not so small when the arguments are fractions.
    * In Eq. 2 $sin(2\pi \frac{kn}{L})$ should instead be $\sin \left(\frac{2\pi kn}{L}\right)$
    * The subscript in FreqTST$_{pretraining}$ should not be italicized, it is a descriptor not a variable.
2. If you want to say a number is in the set of real numbers do not just use and $R$, use the correct notation ($\mathbb{R}$). You can do this using the latex mathbb function.
3. Do you need to define MSE and MAE in equation 8? I think people know what that is.
4. Please check if Eqs. 4 and 5 are correct, should $F$ be $H$ or $G$? If not, what is $F$?

## Figures
1. Figure 3 is hard to understand. To me, all of these values look the same. Only a reader with very good eyes and scale judgment can quantify how different these values are. Please put this information in a table.
2. The captions for all figures and tables are very poor or effectively non-existent. The caption needs to describe in detail what is happening so a reader and look at the figure or table in isolation and understand what it is. For figures with multiple panels, the panels must be labeled and described within the caption.

# Overembelishment and unsupported claims
The vast majority of the claims this work makes and statements about other works are overembellished and are unsupported. Some of which are so egregious that the authors are contradicting themselves. When describing results do not state your own opinion or judgment on the results' quality, state exactly what you observe so that others can make their own judgment of the quality and success. Below are some of the most egregious examples, but this behavior is pervasive.

* "Firstly, our FreqTST outperforms all other supervised baseline models, including PatchTST and other frequency-based models, which verifies the effectiveness of our model"
    * This is one of the primary statements of the paper and comes right after introducing the primary results in Table 2. This statement is completely false! Worse, the evidence of its error is right above it. Between the MSE and the MAE exactly half of the time the pretrained PatchTST outperforms your model. If the PatchTST pretraining is unsupervised this must be made very clear to the reader.

* Experimental results are demonstrated in Figure 4. Both FreqTST and FreqTST$_{pretraining}$ show more promising performances on the zero-shot setting than PatchTST, which even outperform some supervised models (AutoFormer and FEDFormer).
    * Again, in some cases your method outperforms and in other cases it does not. You need to say this specifically! It would be helpful if you said, in words, when it outperformed and by how much and when your model underperformed and by how much. It is better to state your results clearly and it will be apparent to the reader if your model is superior or not. Do not write about your own opinions and judgments.

* "Through comprehensive experiments conducted on multiple datasets, we demonstrate superior performance ..."
    * I would not consider calculating the MSE and MAE on a subset of the datasets in Nie et. al. comprehensive. Why didn't you apply this method to the illness dataset and others? If you had an appendix that showed me more information about your experiment I may consider it comprehensive, but supplying only the MSE and MAE is not comprehensive.
    * Do not claim your model is superior unless it is obvious. Your model is regularly not the best performing model in Tables 2 and 3.

* "The architecture makes full use of the modeling capacity of transformers for discrete and semantic token sequence, and outperforms other supervised models by a large margin"
    * Again, this is not true, in some cases your model outperforms others. When your model does outperform I would not consider this by a large margin. Do not state your own judgments and misrepresent your results.
    * I do not think your model uses the full modeling capacity of transformers (see next section). Please describe why you think this. In general, such a statement is very vague and does not provide insight or understanding of what you are doing.

* In the abstract you say "Transformer-based models have achieved great success in modeling sequence data, especially text, but fail to understand time series sequences. The reason is that individual data points of time series are hard to utilize because they are numerical values that cannot be tokenized."
    * The premise of this paper is that you are going to tokenize these numerical values, but you state that numerical values cannot be tokenized. This contradicts the primary premise of the paper.
   * You state that transformers fail to understand time series sequences. I understand this as saying that the current state of the art fails to understand time series. However, this work claims that the method has an "excellent ability" to perform such tasks and talks about its precision. This might be believable if you substantially outperformed the previous state of the art, but in general you did not outperform it. Either transformers can predict time series or your method also falls short of predicting time series like the other you claim "fail" to do so. The statements in the abstract are not consistent with those later in this work.

* You often use the term *precision* but that is meaningless unless you set a scale. For example, a precision of 1 meter is great if I'm trying to locate a building, but terrible if I am trying to measure the length of a molecular bond. You are saying that this method is in general precise but that is not the case for all problems.


# Methods
## Missing benchmark datasets
The PatchTST (Nie et al., 2023) work shows (Table 3) the MSE and MAE on eight datasets for all the models you mentioned. You need to include the results on these datasets as well for a more comprehensive comparison.

## Fourier transform basis cannot generalize
This work states that the same Fourier dictionary tokenization method can be applied to any other time series. In fact, the zero-shot experiment aims to do just that. This is not entirely true as the number of Fourier coefficients is dependent on the length of the input sequence. If we train a model with $N$ input time points, then it will only expect $N$ unique Fourier coefficients. If one applies this method to a problem where the input sequence is not $N$, the Fourier bases will no longer be orthogonal. This is a drastic change from how the model was originally trained. Applying this model to a problem that has less than $N$ input points means that some of the Fourier coefficients are undefined. When the input sequence is greater than $N$ the Fourier components will not capture the fastest oscillating dynamics within the data and therefore filter out information that might be crucial for the prediction process.

## Misuse of transformer attention mechanism
Attention is meant to highlight similarities between inputs or representations when the ordering or representation of these sequences can change. For example, an attention mechanism will highlight different token positions or have different values for "The ball is blue", "The sphere is blue", and "The blue ball rolls" if asked what color is the ball. Here, Eqs. 4 and 5 determine the embeddings up to a constant so the dot product never changes between their unscaled representations. That is, dot products between $G$ and $H$ never change since they are predetermined. Moreover, the frequency that these bases correspond to will set their order in the frequency series. Therefore, one does not need attention to give us any spacial information since the sequence order in the Fourier domain is set. One only needs to multiply the $a$ and $b$ coefficients together and learn a coefficient to get the values fed into the softmax for attention. Therefore, using attention is a waste of computing time by always recalculating dot products that do not change, instead of multiplying together the Fourier coefficients and learning a weighting coefficient.

## No handling of discrete Fourier transform systematics
The Fourier transform assumes that the dynamics between times [0,T) repeat immediately after. That is, in the range [T, 2T) there is the same signal. This means that the derivatives and values at times 0 and T need to be the same. If not, this introduces systematic errors into the Fourier domain that manifest across the entire spectrum but are most notable at the higher frequencies. This systematic is determined by the difference between the first and last value and their derivatives. Therefore, it changes with each sample and can be confused as a signal. I do not see any handling or mention of this error.

**Questions:**

In the weaknesses section I outlined a number of issues along with examples and, in some cases, ways to make changes. Please address all of them for me to reconsider this work. I have some additional questions below that need to be addressed.

1. Why did the authors choose to represent the Fourier embeddings with Eqs. 4 and 5? This is unclear from the paper. The equations here represent the spatial embedding of a transformer that is added across time, but here they are being used to define the structure of each embedding.
    * Is $F$ supposed to be $G$ and is the second $G$ supposed to be $H$?
2. What is the purpose of $W_\theta$ in Eq. 6? There is nothing special about the imaginary axis. For a Fourier transform the real components represent the even contributions (cosine) and the imaginary components represent the odd (sine) contributions. I do not understand what the complex domain information is that is mentioned.
3. Please evaluate your model on all the datasets in Nie et. al., 2023.

---

> ### Author Response · Authors · 2023-11-23
>
> Thanks for your insightful feedback, which is valuable and very helpful for improving our paper.
>
> Q1: Writing Problems. Thank you very much for your meticulous reading and pointing out the various language errors in the paper. We will modify them and conduct thorough proofreading for our paper.
>
> Q2: Overclaimed problems. (1) We conclude that "Firstly, our FreqTST outperforms all other supervised baseline models, including PatchTST and other frequency-based models, which verifies the effectiveness of our model" means compare FreqTST with other baseline excluding PatchTST with pretraining, which can be understood clearly with the last sentence in implementation details. (2) Figure 4 will be replaced by a more informative table with results on different datasets, which can validate the claim clearly in our paper. (3) The words "comprehensive" and "superior" does overclaim, and we will revise them by "amounts of" and "promising". We will also correct similar errors in the paper, such as instances of "outperform" in certain places. (4) There is probably some misunderstanding in the statement of "they are numerical values that cannot be tokenized", where we mean a single number in time series can not be tokenized as it contains no information alone. However, if a single number represents a frequency of a time series, it obtains semantic information so that it can be tokenized into a vector space. (5) In most scenarios, our method outperforms other baseline by a large margin, for example, in long-term prediction on Etth2 dataset, our method get an impressive results with a average decease of 6%.
>
> Q3: Limited Dataset. To address your concern, we have tried our best to add the required results. The table below presents the experimental results of our model on the Ettm2 dataset.
> |      Models     | Metric |   96  |  192  |  336  |  720  |
> |:-:|:-:|:-:|:-:|:-:|:-:|
> |  FreqTST_ self  |   MSE  | 0.119 | 0.150 | 0.190 | 0.245 |
> |                 |   MAE  | 0.231 | 0.262 | 0.293 | 0.333 |
> |   FreqTST_ sup  |   MSE  | 0.119 | 0.154 | 0.190 | 0.243 |
> |                 |   MAE  | 0.233 | 0.269 | 0.301 | 0.340 |
> | PatchTST_ self_ |   MSE  | 0.120 | 0.147 | 0.180 | 0.238 |
> |                 |   MAE  | 0.231 | 0.260 | 0.286 | 0.331 |
>
> Q4: Linear mapping matrix. We constructed two frequency vector spaces, where G and H represent the basis vector of real and imaginary domains, respectively. After applying FFT to a time series, the coefficients obtained in the real and imaginary domains serve as coordinates for the basis of the two vector spaces. Finally, we use a linear mapping matrix to transform vectors from the imaginary domain to the real domain, and then sum the two, resulting in the final vector in the frequency space. This allows us to represent a time series as vectors in a frequency vector space.
>
> Q5: Learnable embedding. We add learnable embedding when constructing frequency tokens. However, the results are similar to the original version.
> |          Models         | Dataset | Weather |       |       |       | Etth2 |       |       |       |
> |:-----------------------:|:-------:|:-------:|:-----:|:-----:|:-----:|:-----:|:-----:|:-----:|:-----:|
> |                         |  Metric |    96   |  192  |  336  |  720  |   96  |  192  |  336  |  720  |
> |      FreqTST_ self      |   MSE   |  0.178  | 0.223 | 0.281 | 0.356 | 0.184 | 0.220 | 0.250 | 0.313 |
> |                         |   MAE   |  0.220  | 0.256 | 0.299 | 0.347 | 0.291 | 0.319 | 0.344 | 0.390 |
> | FreqTST_ self_learnable |   MSE   | 0.180   | 0.223 | 0.280 | 0.357 | 0.183 | 0.223 | 0.250 | 0.315 |
> |                         |   MAE   | 0.220   | 0.257 | 0.298 | 0.347 | 0.289 | 0.323 | 0.343 | 0.391 |
>
> Q6: Dependency on sequence length. A very insightful observation. Due to the limitations of DFT, we acknowledge that our tokenization method is highly correlated with input length. However, this does not imply weak generalization of our method. On the one hand, upon careful examination of the expressions for G and H, we can find that regardless of changes in input length, as long as the embedding dimensions remain constant, the basis vectors at each position in G and H will not change. This ensures the generalization of token dictionaries. On the other hand, any time series can be partitioned into fixed-length segments. For instance, a time series of length 200 can be divided into 105 subsequences of length 96 (105 = 200 - 96 + 1), and these subsequences can be sampled at regular intervals for prediction. The average results from these predictions can be considered as the final prediction result. This approach allows for extension to time series tasks of arbitrary lengths.

---

### Official Review · Reviewer_je2G · 2023-10-31

**Soundness:** 2 fair
**Presentation:** 3 good
**Contribution:** 1 poor
**Rating:** 3
**Confidence:** 4

**Summary:**

This paper propose to use frequency domain information as tokenizer for time series data, and utilize transformer for further modeling.
The authors also introduce two pretraining tasks based on the proposed model structure.

**Strengths:**

1. This paper proposes to use frequency spectrum for tokenizing time series sequences.
2. This paper designs FreqTST to furthur modeling tokenized time series data and design two pretraining tasks to improve the generalization ability of FreqTST
3. This paper demonstrate that FreqTST achieves promising results on long-term forecasting and zero-short transfer tasks.

**Weaknesses:**

1. The overall novelty of the proposed method is limited. Frequency domain has been widely used in deep learning methods for time series analysis. Although the authors claim they are the first to tokenize continuous time series with frequency domain, there is no fundamental difference between this tokenization method between previous work which utilize FFT to conduct feature extraction.
2. No comparison was made with the latest methods that utilize frequency domain information or employ Transformer structures, e.g., TimesNet and CrossFormer
3. The setting of zero-shot transfer experiment is quite limited. The author only gives results on a set of pre-training and transferred data sets. And the performance of FreqTST for zero-shot learning can be over-claimed.
4. The author doesn't seem to have conducted multiple repeat experiments. The standard deviation of the results is not given in the text, which reduces the persuasiveness of the results.

**Questions:**

1. Why not use cross entropy loss for MPP task, which is more appropriate as it is a classification task.
2. You mentioned FreqTST have lower computation costs than PatchTST, however theoretically, FreqTST applies self-attention on a longer sequence (L / 2) than PatchTST does (L / P). Why PatchTST can not be applied on some of the long-term forecasting tasks while FreqTST could?

---

> ### Author Response · Authors · 2023-11-23
>
> Thanks for your kindly suggestions.
>
> Q1: Noverty of FreqTST. There are some fundamental differences between previous work and FreqTST. Firstly, we propose the concept of frequency token for the first time, which maps the frequency value into a high dimension vector. Secondly, we consider frequency spectrum as a common dictionary for all time series and it can model various time series in a unfied way, while previous work only extract features on single dataset. Thirdly, all previous frequency-based methods transform signals into temporal domain after extracting features in frequency domain. At the same time, FreqTST uses frequency-tokenised representation to predict future values directly, filling the gap between frequency domain and temporal domain.
>
> Q2: Comparison with latest model. PatchTST(ICLR 2023) and DLinear(AAAI 2023) are contemporaneous works with Timesnet(ICLR 2023) and Crossformer(ICLR 2023), so the first two models involved in our paper can serve as representatives of the latter two models.
>
> Q3: Completion of zero-shot experiments. It is very kind of you to point out this problem. In fact, we conduct extra experiments but only show one sample in our paper. Here are more results that show the powerful transferibility of FreqTST.
> |        Models       | Dataset | Weather |       |       |       | Ettm2 |       |       |       | Etth2 |       |       |       |
> |:--:|:-:|:--:|:-:|:-:|:-:|:-:|:-:|:-:|:-:|:-:|:-:|:-:|:-:|
> |                     |  Metric |    96   |  192  |  336  |  720  |   96  |  192  |  336  |  720  |   96  |  192  |  336  |  720  |
> |  PatchTST_ sup_zero |   MSE   | 0.232   | 0.276 | 0.338 | 0.418 | 0.173 | 0.193 | 0.236 | 0.286 | 0.201 | 0.243 | 0.275 | 0.354 |
> |                     |   MAE   | 0.273   | 0.306 | 0.350 | 0.402 | 0.294 | 0.309 | 0.342 | 0.373 | 0.302 | 0.333 | 0.359 | 0.416 |
> | PatchTST_ self_zero |   MSE   | 0.234   | 0.280 | 0.329 | 0.397 | 0.170 | 0.190 | 0.220 | 0.275 | 0.248 | 0.276 | 0.304 | 0.360 |
> |                     |   MAE   | 0.281   | 0.314 | 0.344 | 0.385 | 0.290 | 0.304 | 0.325 | 0.361 | 0.348 | 0.366 | 0.384 | 0.420 |
> |  FreqTST_ sup_zero  |   MSE   | 0.204   | 0.249 | 0.304 | 0.392 | 0.177 | 0.206 | 0.276 | 0.405 | 0.203 | 0.237 | 0.271 | 0.361 |
> |                     |   MAE   | 0.258   | 0.293 | 0.334 | 0.396 | 0.297 | 0.322 | 0.375 | 0.455 | 0.305 | 0.334 | 0.366 | 0.432 |
> |  FreqTST_ self_zero |   MSE   | 0.247   | 0.264 | 0.328 | 0.373 | 0.142 | 0.183 | 0.205 | 0.258 | 0.209 | 0.235 | 0.264 | 0.333 |
> |                     |   MAE   | 0.271   | 0.290 | 0.335 | 0.362 | 0.257 | 0.291 | 0.310 | 0.346 | 0.312 | 0.330 | 0.352 | 0.402 |
>
> Q4: Multiple repeat Experiments. In the field of deep learning, repeated experiments are indeed essential. However, many time-series-related works presented at ICLR 2023 (Timesnet, Crossformer) did also not report the results of repeated experiments and their standard deviation. This is because the experimental results in this field are relatively stable with strong reproducibility. Below is a set of repeated experiments from our experimental process, showing that there is almost no difference in the results at this level of precision.
> | weather |   96  |  192  |  336  |  720  |
> |:-:|:-:|:-:|:-:|:-:|
> | result1 | 0.175 | 0.221 | 0.277 | 0.356 |
> |         | 0.215 | 0.255 | 0.296 | 0.347 |
> | result2 | 0.174 | 0.221 | 0.277 | 0.355 |
> |         | 0.214 | 0.255 | 0.296 | 0.347 |
>
> Q5: Cross entropy for MPP task. Thanks for your valuable suggestion. We conducted validation experiments using the cross entropy loss and found that the results were not as good as those using MSE. We think this is because the pre-training loss function is the sum of the loss functions from the two pre-training tasks. If the loss functions of the two pre-training tasks are different, the optimization space of the entire loss function will be more rugged, making it difficult to converge to the optimal solution.
> |        Models       | Dataset | Weather |       |       |       | Etth2 |       |       |       |
> |:-:|:-:|:-:|:-:|:-:|:-:|:-:|:-:|:-:|:-:|
> |                     |  Metric |    96   |  192  |  336  |  720  |   96  |  192  |  336  |  720  |
> |     FreqTST_ sup    |   MSE   |  0.169  | 0.210 | 0.258 | 0.326 | 0.185 | 0.225 | 0.236 | 0.291 |
> |                     |   MAE   |  0.229  | 0.267 | 0.307 | 0.358 | 0.294 | 0.327 | 0.338 | 0.381 |
> |    FreqTST_ self    |   MSE   |  0.178  | 0.223 | 0.281 | 0.356 | 0.184 | 0.220 | 0.250 | 0.313 |
> |                     |   MAE   |  0.220  | 0.256 | 0.299 | 0.347 | 0.291 | 0.319 | 0.344 | 0.390 |
> | FreqTST_ self_cross |   MSE   |  0.183  | 0.223 | 0.279 | 0.356 | 0.181 | 0.220 | 0.253 | 0.321 |
> |                     |   MAE   |  0.223  | 0.256 | 0.297 | 0.348 | 0.287 | 0.319 | 0.346 | 0.394 |
>
> Q6: Long-term prediction of PatchTST. $PatchTST_{pretraining}$ can not afford to predict long-term time series as it has a huge linear layer. However, FreqTST in supervised costs less in memory.

---

### Official Review · Reviewer_qa3G · 2023-11-10

**Soundness:** 2 fair
**Presentation:** 2 fair
**Contribution:** 3 good
**Rating:** 5
**Confidence:** 4

**Summary:**

This paper studies time series forecasting using transformers that processes the time series in the frequency domain. For this a vocabulary of frequency tokens are created to represent time series in terms of frequency tokens. The frequency tokens are used as input to the transformers model. For training the transformer model two pertaining tasks used: i) Time series reconstruction from the frequency domain representation, and ii) maximum position prediction task. The model is evaluated on supervised training setup and transfer learning set up. For the transfer learning setup, the final model is fine-tuned on a large scale time series dataset and applied on other time series predictions. The proposed model achieves comparable results with PatchTST.

**Strengths:**

* Creating vocabulary of frequencies interesting way of presenting the frequency domain vectorisation.
* The proposed model obtains comparable results with PatchTST
* The pre-taining tasks sound reasonable

**Weaknesses:**

* The overall paper is well-written.
* The motivation/need of a discrete tokenised representation is not clear for time series forecasting. Especially the formulation of pre-training not in the form of MLM indicates that it doesn’t necessarily builds on contextualised formulation.
* There are some issues as over-useage of notation in the methodology section: In Equation 2 i is used for imaginary part, but in Equation 4 and 5 i is used for index of the position.
* Some experimental details are missing, e.g. It is mentioned beta value is altered between different values.
* Some work on frequency domain forecasting is not mentioned in the related work
     * Chen et al. (2023) FrAug: Frequency Domain Augmentation for Time Series Forecasting
     * Sun et al. (2022) FreDo: Frequency Domain-based Long-Term Time Series Forecasting

**Questions:**

* It is not clear why the data augmentation is only applied for frequency domain models. Could you explain what is rationale behind this?
* Did you apply data augmentation for the other models?
* Did you try pre-training with MLM like loss, how did it behave?
* How the beta value alters, is there a scheduling used for it?
* Is multi-head attention used? What is head size? What is the embedding size and hidden size for attention layers? Which activation function is used? Were there any drop out? Which optimization is used? What is the optimization hyperparameters? How the learning rate? * How the baseline models trained?
* How does the model perform on other common time series datasets like traffic dataset that is also used in PatchTST paper?

---

> ### Author Response · Authors · 2023-11-23
>
> Thanks for your recognition and kindly comments.
>
> Q1: Motivation of discrete tokenised representation. We discussed our motivation in Sec 1. On the one hand, discrete tokenised representation has shown powerful capability to model text data, which is worthy of reference in the time series domain. On the other hand, it is an effective way to model different kinds of time series using a unified discrete tokenised representation, which is beneficial to train a large model of time series on various datasets.
>
> Q2: MLM pretraining task. We have considered the issue once we designed this method. Here are the reasons that we do not use MLM task. Firstly, considering that the downstream task involves predicting time series, if the pre-training process focuses solely on correlation calculations in the frequency domain using MLM task, the model may struggle to grasp changes in the time series effectively. Secondly, the frequency basis vectors are independent from each other. Attempting to reconstruct missing vectors based on other representation vectors is difficult. Thirdly, the results below indicate that the MLM task is not suitable for this particular scenario.
> | Dataset | TSR | MPP | MLM | Metric | 96    | 192   | 336   | 720   |
> |-|-|-|-|-|-|-|-|-|
> | Weather |     |     | √   | MSE    | 0.179 | 0.230 | 0.282 | 0.358 |
> |         | √   |     |     | MSE    | 0.174 | 0.221 | 0.276 | 0.354 |
> |         |     | √   |     | MSE    | 0.176 | 0.220 | 0.276 | 0.354 |
> |         | √   | √   |     | MSE    | 0.173 | 0.220 | 0.276 | 0.353 |
>
> Q3: Notation issue: Thank you very much for your meticulous reading and pointing out the notation issues in the paper. We will modify them and conduct thorough proofreading for our paper.
> Q4: Beta value: We have conducted a considerable number of experiments to evaluate the effect of beta value. It concludes that it has little influence on the final results and the best results are achieved when beta is set to 1.
> | Model                  | Metric | 96    | 192   | 336   | 720   |
> |-|-|-|-|-|-|
> |   FreqTST_ self_beta1  |   MSE  | 0.178 | 0.223 | 0.281 | 0.356 |
> |                        |   MAE  | 0.220 | 0.256 | 0.299 | 0.347 |
> |  FreqTST_ self_beta0.5 |   MSE  | 0.178 | 0.225 | 0.279 | 0.357 |
> |                        |   MAE  | 0.219 | 0.258 | 0.298 | 0.348 |
> | FreqTST_ self_beta0.75 |   MSE  | 0.178 | 0.225 | 0.280 | 0.356 |
> |                        |   MAE  | 0.218 | 0.259 | 0.298 | 0.348 |
> |  FreqTST_ self_beta1.5 |   MSE  | 0.182 | 0.223 | 0.280 | 0.356 |
> |                        |   MAE  | 0.222 | 0.256 | 0.297 | 0.347 |
> |   FreqTST_ self_beta5  |   MSE  | 0.177 | 0.224 | 0.281 | 0.356 |
> |                        |   MAE  | 0.217 | 0.258 | 0.299 | 0.347 |
>
>
> Q5: Relevant research. It is very kind of you for recommending papers highly related to our work. The Fredo model concludes that learning in frequency domain improves model performance with statistical significance, which is a great support material for our work. At the same time, we would like to emphasize that we cited FrAug in Sec 3.1 instead of in related work, as it mainly provides a method of data augmentation rather than model designing.
>
> Q6: Data augmentation: We only did the data augmentation on frequency-based model because it mainly involved frequency computation in the method of FrAug. Moreover, the results also illustrate that it exhibits non-frequency-based models achieve similar performance whether utilizing the FrAug method or not.
>
> |    Models   | Metric |   96  |  192  |  336  |  720  |
> |:-:|:-:|:-:|:-:|:-:|:-:|
> |   DLinear   |   MSE  | 0.195 | 0.236 | 0.283 | 0.352 |
> |             |   MAE  | 0.253 | 0.294 | 0.333 | 0.391 |
> | DLinear_aug |   MSE  | 0.196 | 0.236 | 0.286 | 0.346 |
> |             |   MAE  | 0.253 | 0.291 | 0.337 | 0.383 |
>
> Q7: Experimental details. We apologize for not clarifying the detailed issues, and draw a table in appendix for all the parameters you mentioned. Note that we use a learning rate schedule the same as PatchTST, which is altered according to different datasets. The training settings of baseline models are consistent with the original paper.
> | parameter | head size | embedding size | hidden size | dropout | optimizer | activation function |
> |:-:|:-:|:---:|:--:|:---:|------|---------|
> |   value   |     16    |       128      |     256     |    0    | Adam      | gelu                |
>
> Q8: Performance on other dataset. To address your concern, we add the result below.
> |       Models       | Metric |   96  |  192  |  336  |  720  |
> |:------------------:|:------:|:-----:|:-----:|:-----:|:-----:|
> |    PatchTST_self   |   MSE  | 0.510 | 0.506 | 0.507 |   -   |
> |                    |   MAE  | 0.329 | 0.324 | 0.319 |   -   |
> | PatchTST_supervise |   MSE  | 0.446 | 0.453 | 0.468 | 0.501 |
> |                    |   MAE  | 0.283 | 0.286 | 0.291 | 0.310 |
> |    FreqTST_self    |   MSE  | 0.542 | 0.533 | 0.552 | 0.593 |
> |                    |   MAE  | 0.335 | 0.328 | 0.333 | 0.351 |

---

### Meta-Review · Area_Chair_f5Dm · 2023-12-06

**Metareview:**

This paper presents a Transformer-based model for time series forecasting, utilizing an innovative tokenization scheme for continuous valued time-series data. All reviewers agree on the relevance and interest of the issue addressed. The proposed tokenization method is interesting, but the novelty is limited. The authors missed to discuss several related works. Moreover, while and the model's performance is on par with PatchTST, a detailed comparison with other state-of-the-art methods is missing. While one reviewer showed support, two others expressed significant concerns. Key issues include (i) soundness of the technical section; (ii) quality of presentation; (iii) novelty of the proposed approach; (iv) limited experimental evaluation. The authors provided additional results during the rebuttal phase, which do help to improve the paper. However, the number of issues and questions raised by the reviewers suggest that major revisions are necessary. The paper, in its current form, is not ready for publication. Therefore, I recommend rejection of this submission.

**Justification For Why Not Higher Score:**

The paper in its current form is not ready for submission. The rebuttal phase could not resolve all the issues.

**Justification For Why Not Lower Score:**

N/A

---

### Decision · Program_Chairs · 2024-01-16

Reject